

# Review of knockout technology approaches in bacterial drug resistance research

Chunyu Tong*, Yimin Liang*, Zhelin Zhang, Sen Wang, Xiaohui Zheng, Qi Liu and Bocui Song

College of Life Science and Technology, Heilongjiang Bayi Agricultural University, Daqing, Heilongjiang, China
* These authors contributed equally to this work.

## ABSTRACT

Gene knockout is a widely used method in biology for investigating gene function. Several technologies are available for gene knockout, including zinc-finger nuclease technology (ZFN), suicide plasmid vector systems, transcription activator-like effector protein nuclease technology (TALEN), Red homologous recombination technology, CRISPR/Cas, and others. Of these, Red homologous recombination technology, CRISPR/Cas9 technology, and suicide plasmid vector systems have been the most extensively used for knocking out bacterial drug resistance genes. These three technologies have been shown to yield significant results in researching bacterial gene functions in numerous studies. This study provides an overview of current gene knockout methods that are effective for genetic drug resistance testing in bacteria. The study aims to serve as a reference for selecting appropriate techniques.

## INTRODUCTION

Microbial drug resistance has become a significant challenge in preventing and controlling microbial infections (*Larsson & Flach, 2022*). Bacteria develop multi-drug resistance through three key mechanisms: overexpression of genes linked to active drug efflux pump, reduced extracellular drug uptake due to decreased expression of outer membrane proteins, and inactivation of antibiotics through the production of intracellular antibiotic hydrolases (*Ghosh et al., 2020*; *Yu et al., 2020*). These mechanisms allow bacteria to regulate their internal environment and protect themselves from harm. The development of sequencing technology and bioinformatics have resulted in the widespread use of high-throughput sequencing techniques for detecting bacterial drug resistance. These techniques include whole genome sequencing (WGS), metagenomics sequencing (mNGS), and microbial-single-cell transcriptome sequencing (MscRNA-seq) (*Wang et al., 2020*; *Tang et al., 2019*). By utilizing these technologies, bacterial drug resistance genotypes can be identified, new drug resistance genes can be discovered, and transcriptional profiles of drug resistance genes in individual bacterial cells can be obtained (*Zlamal et al., 2021*). The study of gene functions can help in understanding the mechanisms of drug resistance.

Corresponding authors
Chunyu Tong,
Tongchunyu@126.com
Bocui Song, songbocui66@163.com

Therefore, the validation of bacterial gene functions through knockout technology has been extensively researched. This article provides a review of current effective knockout methods for studying the function of bacterial drug resistance genes. Knockout eliminates the function of a drug resistance gene, while gene overexpression can positively verify the gene's function. Gene complementation can theoretically revert the changes of knockout. By constructing knockout, complementation, and overexpression strains and comparing and assessing the differences between them, researchers can precisely explore the function of the target gene. The aim of this review is to provide relevant researchers with a reference for selecting knockout techniques. Portions of this text were previously published as part of a preprint (*Tong et al., 2023*).

## SURVEY METHODOLOGY

To identify articles related to bacterial drug resistance, we conducted a search on the Pubmed database using the keywords 'gene knockout' and 'bacterial drug resistance'. We then selected articles that matched the topic based on their titles and abstracts. To further refine our search, we added 'λ-Red homologous recombination', 'CRISPR/ Cas9', and 'suicide plasmid' as additional keywords for the three main topics in the article. When searching for articles, it is important to carefully read and filter out those that are relevant and well-represented, while also ensuring that references are current.

### Common bacterial gene knockout techniques

Homologous recombination is a widespread physiological phenomenon in living organisms that serves as an intrinsic mechanism for correcting DNA mutations induced by external factors or internal processes (*Sun et al., 2020*). This molecular biological process is the basis for gene knockout. There are three types of knockout technologies developed based on this process. The first is the Red recombination system, which utilizes the Red region of the λ phage gene to encode a gene capable of initiating homologous recombination of bacterial chromosomes with exogenous DNA, specifically for prokaryotic cells (*Zhang et al., 2000*). Two common methods for knocking out specific genes in eukaryotic cells are the Cre-LoxP and Flp-FRT systems (*Djukanovic et al., 2006*). These systems involve a specific DNA sequence and a recombinase to target the gene of interest. Another method for achieving genome-wide knockout of any gene is through large-scale random insertion mutations, such as transposons and T-DNA insertion mutations (*McClintock, 1950*). These insertion mutations have become the more effective methods currently used in plants.

The efficient suicide vector system is a chromosome modification technology that was developed in the 1980s. Its primary objective is to construct recombinant suicide plasmids that contain homologous DNA fragments of a certain length. The host's recombination system is used for the exchange between homologous recombination sequences The suicide vectors can knock out target genes and eliminate exogenous fragments after two homologous recombination processes (*Jain & Ertesvåg, 2022*; *Wang et al., 2019*). This system has a broad host range and transferability, making it useful for studying bacterial physiology, virulence, and drug-resistance genes.
According to *Rudin, Sugarman & Haber (1989)* and *Rouet, Smih & Jasin (1994)*, the introduction of double-strand break (DSB) DNA at the target site significantly improves knockout efficiency. Homing nucleases, which are extensive nucleic acid endonucleases, are used to introduce specific double-strand break DNA into the genome, mostly through non-homologous end joining (NHEJ) repair. Based on this method, genome editing is primarily carried out using the TALEN, ZFN, and CRISPR/Cas systems.

The use of ZFN and TALEN gene editing technologies has been widely adopted for genome editing in eukaryotic organisms, such as animals and plants. These technologies allow for precise genome manipulation, including insertion, deletion, or replacement, and have been found to be more efficient than traditional homologous recombination methods (*Zhang, Zhang & Yin, 2019*). However, their complexity, long cycle time, and high cost are considered to be their main disadvantages (*Cui et al., 2021*). While ZFN and TALEN technologies for bacterial gene editing have been rarely reported, third-generation CRISPR/Cas9 gene editing has emerged as a more efficient, faster, and easier alternative to the previous two generations. However, off-target effects remain a significant concern (*Corsi et al., 2022*). CRISPR/Cas9 has been extensively utilized for gene editing in both eukaryotes and prokaryotes, including *Escherichia coli* (*E. coli*) (*Pyne et al., 2015*; *Jiang et al., 2015*).

Red homologous recombination, CRISPR/Cas9, and suicide plasmid techniques are commonly used for gene knockout in the laboratory. All three methods induce DNA double-strand breaks to replace the target gene through the non-homologous end or homologous end repair mechanisms. The main strategies are elucidated as follows (*Muyrers, Zhang & Stewart, 2001*): (1) One-step screening, with a screening marker in the middle and linear DNA fragments with homologous sequences at both ends of the gene. The marker gene displaces the target gene with the help of recombination between the homologous sequence and the target vector. (2) In the recombination process, a linear DNA fragment with two screening genes and a reverse screening gene is used. The first round of recombination selects the recombinant molecule using the screening marker gene. In the second round, the homologous fragment displaces the two screening genes on the recombinant molecule, and the reverse screening marker gene is used for selection. The screening marker genes contain specific sites recognized by Cre or FLP site-specific recombinase, which can be deleted after the first screening, leaving a special sequence of more than ten base pairs on the recombinant molecule. (4) Restriction sites were located on both sides of the screening marker genes, allowing for cleavage by restriction endonucleases and subsequent ligation of recombinant molecules. The combination of Red homologous recombination technology with other DNA experimental techniques has greatly expanded the technical means of gene manipulation (*D'Souza et al., 2021*; *Sugawara et al., 2022*; *Liu et al., 2020a*). Researchers have the freedom to utilize these techniques to achieve various experimental objectives.

## Common knockout techniques in bacterial drug resistance research
### Red homologous recombination technologies

Knockout procedures in certain Gram-negative bacteria, such as *E. coli*, *Salmonella*, and *Klebsiella*, typically involve a two-step homologous recombination approach. The RecA and RecBCD proteins encoded by these strains are usually utilized as mediators in the conventional method, which has been studied extensively over time (*Wiktor et al., 2021*). Although it is established that Gam proteins can bind to RecBCD exonuclease and inhibit nucleic acid exonuclease activity, the exact mechanism behind this process is still unclear (*Chen et al., 2016*). As a result, obtaining the desired recombinant can be challenging, leading to reduced knockout efficiency and hindering the development of this technique.

*Zhang et al. (1998)* proposed that the RedE/RecT system in *E. coli* demonstrates a recombination function. It was discovered that the Exo and Beta proteins of λ-phage also exhibit the same function (*Zhang et al., 2000*). The process was later termed ET Recombination and can significantly improve the conventional method's shortcomings by reducing the required homologous arms and increasing the recombination rate (*Rivero-Müller, Lajić & Huhtaniemi, 2007*). As a result, it has been widely used in the genetic modification of *E. coli*.

The process of recombining double-stranded DNA (dsDNA) requires three specific bacteriophage λ-Red proteins namely Gam, Exo, and Beta (*Yu et al., 2003*). Exo proteins are exonucleases which degrade dsDNA from the 5' end. On the other hand, Beta proteins bind the single-stranded regions that are derived from Exo proteins and facilitate recombination by promoting pairing with cognate genomic targets (*Sawitzke et al., 2007*). Lastly, Gam prevents the degradation of linear dsDNA by *E. coli* RecBCD protein and SbcD nuclease (*Murphy, 2012*).

*Muyrers et al. (1999)* initially constructed a plasmid expressing phage recombinase. Since then, this plasmid has been modified by numerous researchers. The most widely used version is pKD46 developed by *Datsenko & Wanner (2000)*. This plasmid can express the complete λ-Red proteins in the auxotrophic plasmid pKD46 and has revolutionized the two-step homologous recombination method. It allows for successful gene knockout in *E. coli* K-12. The Red recombinant technology established using pKD46 requires only 36 nucleotides for the homologous arm in the primer. The *exo*, *bet*, and *gam* genes are controlled by the arabinose promoter on the temperature-sensitive plasmid pKD46. The DNA fragment used to replace targeted gene can be obtained from the pKD3 and pKD4 plasmids, which contain chloramphenicol and kanamycin resistance genes, respectively, along FRT sites (flippase recognition target) (*Broach & Hicks, 1980*). The resistance gene can be removed with the help of pCP20 plasmid, which expresses FLP recombinase that recognizes direct repeat FRT sites flanking the resistance gene (as shown in Fig. 1, adapted from (*Datsenko & Wanner, 2000*)). The temperature-sensitive replicons present in pKD46 and pCP20 plasmids can be eliminated by incubating them at 37 °C.

*Feng et al. (2020)* utilized the Red recombinant system to create a gene deletion strain of the *E. coli* efflux protein YddA, AcrB, and then the drug susceptibility of the deletion strain. Their findings suggest that YddA functions as an ATP-dependent efflux protein that

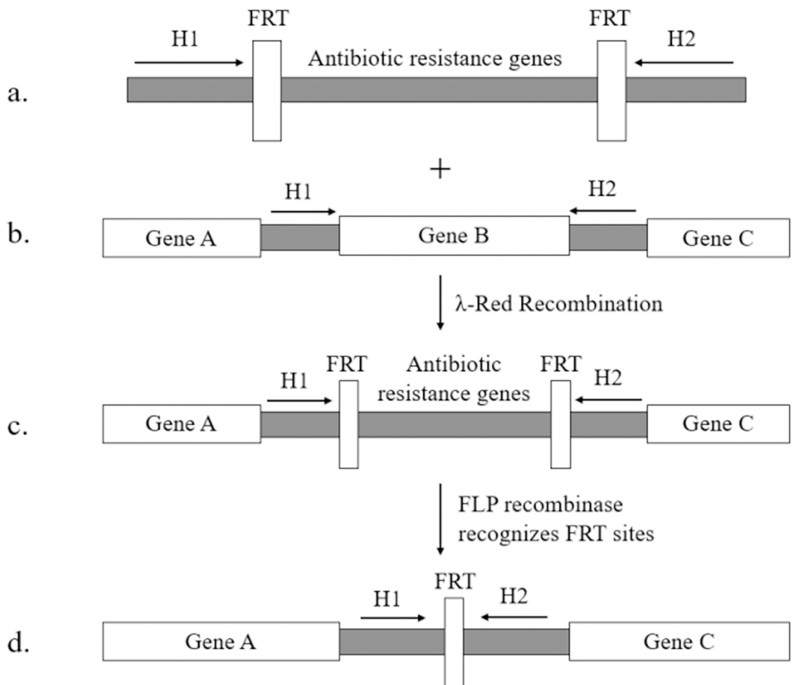

**Figure 1 λ-Red homologous recombination.** (A) The targeting fragment contains homologous arms (H1, H2), FRT sites and resistance genes for screening; (B) the homologous arms (H1, H2) on both sides of the target gene are used to undergo homologous recombination; (C) the targeting fragment successfully displaces the target gene under the mediation of Red homologous recombination; (D) the pcp20 plasmid expresses FLP recombinase, recognizes the FRT sites and eliminates the resistance genes.

mainly effluxes norfloxacin. *Ogawa et al. (2015)* investigate the mechanism of multidrug and toxic compound extrusion (MATE) resistance of *Klebsiella* using the same recombinant plasmid. Their results indicate knocking down the *ketM* gene of the exocytosis pump does not change that the MIC values of antibacterial drugs such as kanamycin. Therefore, it confirms that the MATE exocytosis pump is not the direct cause of drug resistance in *Klebsiella*.

## CRISPR/Cas9 technologies

CRISPR technology was discovered in 1987. In 2014, Doudna and Charpentier confirmed *in vitro* experiments that the CRISPR-cas9 system can 'localize' DNA breaks (*Doudna & Charpentier, 2014*). The CRISPR knockout is used in functional genomics studies to detect genomic loci of cellular drug resistance (*Hilton & Gersbach, 2015*; *Koike-Yusa et al., 2014*; *Zhou et al., 2014*), elucidate how cells induce host immune responses, and determine how certain viruses cause cell death (*Ma et al., 2015*). This technique has been widely applied in both prokaryotes and eukaryotes.

CRISPR/Cas systems are divided into two categories: Class 1 and Class 2. Types I, III, and IV are part of Class 1, while II, V, and VI belong to Class 2 CRISPR/Cas systems (*Koonin & Makarova, 2019*). Class 1 systems share some common features, such as the processing of precursor crRNAs using specialized Cas endonucleases. Once mature, the
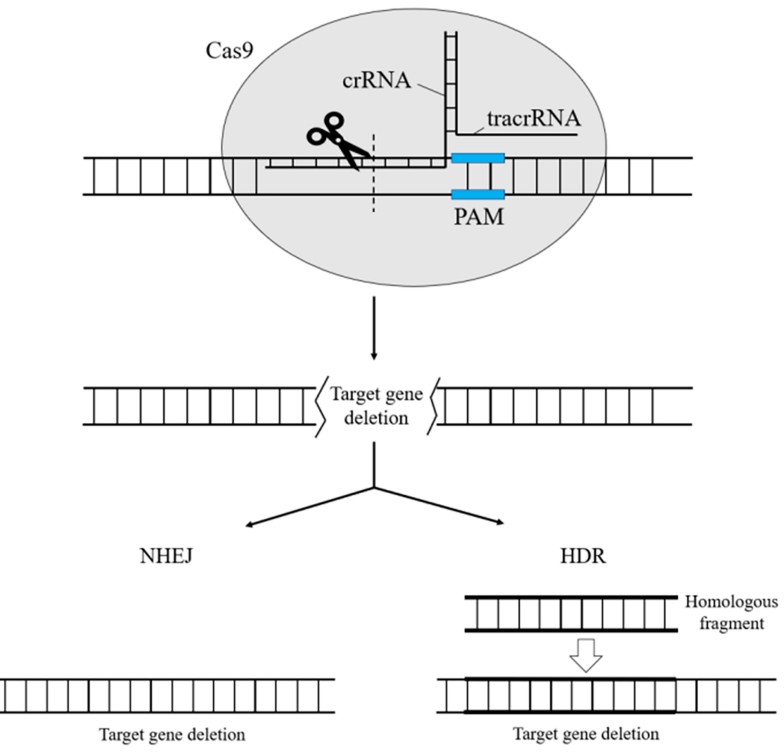

**Figure 2 CRISPR/Cas9 for gene knockout.** crRNA binds to tracrRNA through base pairing to form Guide RNA, which forms a complex with the nuclease Cas9 and guides Cas9 to shear ds DNA at the sequence target site paired with crRNA. After shearing down the target gene, the cell can use its own non-homologous end-joining repair to repair the broken chromosome, or it can use Red homologous recombination, which artificially introduces homologous fragments for homologous end-joining repair.

respective crRNA is assembled into a large multi-Cas protein complex that can recognize and cleave nucleic acids complementary to the crRNA. On the other hand, in Class 2 systems, a single multinomial large protein serves as the effector. In the CRISPR Type II (CRISPR/Cas9) system of *Streptococcus*, the CRISPR-RNA (tracrRNA) is connected to a small complementary crRNA region, forming partial dsRNA. This dsRNA can then bind to Cas9 and target the prototypical spacer sequence, which is subsequently degraded by the nucleic acid endonuclease Cas9 (*Garcia-Robledo, Barrera & Tobón, 2020*) (Fig. 2).

In *E. coli*, artificial double-strand breaks are typically repaired through RecA-mediated homologous recombination using homologous sequences as editing templates. However, repairing double-strand breaks caused by CRISPR-Cas9 cutting of chromosomes is generally challenging for natural homologous recombination pathways. The introduction of phage-derived λ-Red recombinase into the CRISPR system improves the likelihood of obtaining mutant strains, and precise gene modification can be achieved by supplementing target fragments with CRISPR-Cas9 plasmids (*Jiang et al., 2013*; *Li et al., 2015*; *Pines et al., 2015*). The efficiency of this process can be further increased by introducing an exogenous DNA repair system, such as when target fragments and λ-Red systems are introduced simultaneously (*Huang et al., 2019*).

*Liu et al. (2020b)* demonstrated the effectiveness of the CRISPR-Cas9 system against resistant plasmids by targeting and disrupting plasmids encoding kanamycin resistance genes in *E. coli*, resulting in over 99% of bacteria restoring kanamycin susceptibility for 32 h. Meanwhile, *Wu et al. (2019)* utilized the CRISPR/Cas9 genome editing function for modification and removal by introducing a repair template for homologous recombination, which significantly increased gene editing efficiency. Additionally, *Liu et al. (2014)* showed that the CRISPR-Cas9 system can create a *gyrA* gene mutation, altering the susceptibility of the strain to quinolones, thus establishing a causal relationship between the *E. coli gyrA* mutation and its resistance to quinolone antimicrobials.

## Suicide plasmid vector system

The suicide plasmid vector system is a technique developed in the 1980s for modifying chromosomes (*Dower, Miller & Ragsdale, 1988*). It has a wide host range and can be transferable between bacterial cells. These plasmids contain the R6K replication initiator, which only replicates in recipient bacteria with the *pir* gene. Once the *pir* gene is missing, the R6K initiation function is eliminated (*Riedel et al., 2013*; *Penfold & Pemberton, 1992*), making it unable to replicate in general bacteria. Suicide plasmids must be incorporated into the bacterial chromosome or plasmids that replicate along with bacterial proliferation, which is a characteristic condition and special feature of these plasmids.

Suicide plasmids have two options upon transfer into a host cell, as they lack replicon structures that enable replication within the host cell. The first option is automatic removal without replication, while the second is incorporation into the chromosome and replication alongside it. The homologous arm is incorporated into the suicide plasmid vector by enzymatic cleavage and DNA ligation. This results in integration of the homologous fragment from the suicide plasmid with the corresponding fragment on the bacterial genome. This integration allows for accurate gene deletion through the RecA recombination system of the bacteria upon introduction of the suicide plasmid (*Wiktor et al., 2021*).

Gene deletion using suicide plasmids involves a two-step homologous recombination process with a counter-selection (*Caldwell & Bell, 2019*). The first step involves replacing the bacterial target gene with the homologous arm in the suicide plasmid, which integrates the plasmid into the genome. The second step involves removing the plasmid backbone from the bacterial genome to obtain a trace-free gene knockout. The suicide plasmid is then removed by rejection due to its inability to replicate in an environment lacking relevant replication conditions, resulting in a trace-free knockout deletion strain (*Wang et al., 2019*) (Fig. 3).

Two studies have described the use of suicide plasmids to knock out drug resistance genes in bacteria. *Oh et al. (2015)* used overlapping PCR to amplify fragments with the upstream and downstream regions of the target gene and antibiotic resistance genes, which were then ligated into *Acinetobacter baumannii* (*A. baumannii*) for homologous recombination *via* flat ends. *Tsai et al. (2011)* employed the suicide plasmid allele exchange technique to remove the ompK35 from the *Klebsiella* chromosome and reintroduce the ompK36 gene. In comparison to the parental strain, only the strain without ompK36

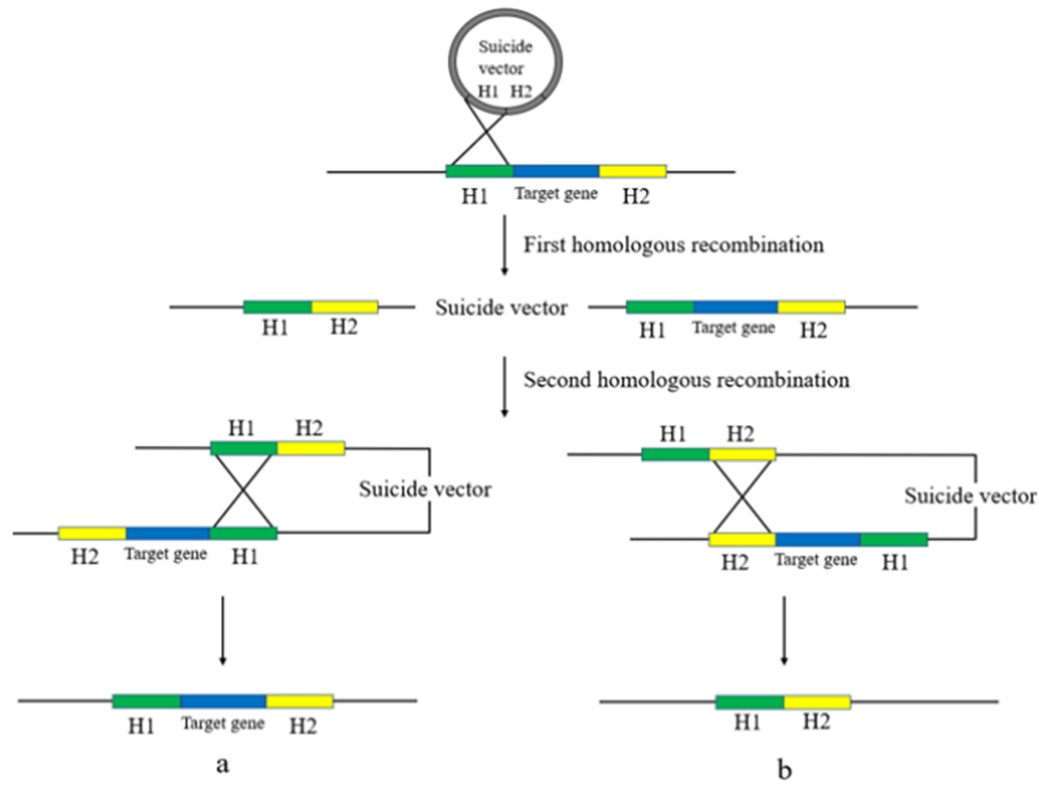

**Figure 3 Suicide plasmid for gene knockout.** Suicide vectors were integrated into the bacterial genome after undergoing the first homologous recombination, after which they underwent the second homologous recombination to obtain two results, strains with the same sequence as the wild type (A), and strains without trace knockout (B).

demonstrated resistance to cefazolin, cephalothin, and cefoxitin. The deletion of ompK35 resulted in even higher MICs, indicating that strains with both deletions were significantly resistant to the aforementioned drugs.

## Advantages and disadvantages analysis

The two-step homologous recombination method involves transferring the plasmid encoding the recombinase into the bacteria first, followed by electroporation of exogenous DNA for homologous recombination. The defects are elucidated as follows: (1) it requires a high concentration of DNA fragments, particularly for strains with dense cell membranes. Due to low uptake efficiency, the amount of DNA entering the cells often fails to meet the minimum standard for homologous recombination, resulting in gene knockout failure. Moreover, the electro-transformation will cause the death of numerous cells, and the remaining cells will not reach the required concentration, thus resulting in knockout failure. (2) The Red system is used to remove the recombinase expression vector from the bacteria, enabling the introduction of another vector expressing the FLP recombinase. This process is repeated to eventually produce a markerless mutant. However, the λ-Red recombination-based approach is time-consuming, and the knockout by Red homologous recombination leaves a residual FRT locus on the bacterial genome.

*Herring, Glasner & Blattner (2003)* introduced the gene gorging method as an alternative to conventional methods. The method involves ligating the targeting fragment onto a plasmid, which allows for mass replication in the host without exogenous transformation. Homing endonuclease cleavage then produces targeting DNA in each cell, increasing the number of cells that may undergo homologous recombination. The single plasmid knockout method proposed by *Yu et al. (2008)*, combines the λ-Red recombination system and homing endonuclease on a plasmid control led by arabinose and rhamnose promoters. The homologously recombined target fragment is inserted into the homing endonuclease for cleavage using double-swap homologous recombination. This process is similar to the suicide plasmid vector system and allows for a trace-free knockout without the need for repeated plasmid transformations. This method is superior to previous techniques and can be used for a variety of gene modification work, such as point mutation of genes in *E. coli* and gene-targeted insertion.

The CRISPR/Cas9 technology has the capability of targeting almost any gene for a traceless knockout with high efficiency. This is achieved by designing a single guide RNA that guides Cas9 protein to track close to the protospacer adjacent motif (PAM) sequence and shear the exogenous dsDNA. However, for most existing industrial microorganisms, non-homologous end recombination is usually preferred for DNA repair. To increase the chance of homologous recombination, *Jiang et al. (2015)* coupled CRISPR/Cas9 technology with Red recombinase, which yielded better results compared to λ-Red recombination technology. Although CRISPR technology has been widely used in eukaryotes, its application in bacteria is less prevalent. Most CRISPR/Cas9 gene editing systems are used on microorganisms to obtain gene deletion strains for further studies. However, these systems have several defects such as off-target effects, unstable tool plasmids, and toxic effects of Cas9 proteins (*Cullot et al., 2019*; *Álvarez, Biayna & Supek, 2022*). Additionally, the targeting efficiency of the Cas9 protein varies in different strains, so it is important to consider using tissue-specific promoters to drive the targeted expression of Cas9 nucleic acid endonuclease in vector plasmid design (*Hsu, Lander & Zhang, 2014*).

Suicide plasmid homologous recombination technology can be used for gene editing, allowing for traceless knockout of target genes. This is accomplished by constructing homologous fragments upstream and downstream of the target gene, which is linked to the suicide plasmid. Bacteria can integrate the suicide plasmid into the chromosome through the RecA recombination function, and then eliminate the plasmid backbone through a second homologous recombination process. One limitation of the aforementioned method is that the suicide plasmid must be transferred to the host bacterium after the conjugational transfer, which introduces some uncertainties during the process. However, since the suicide plasmid cannot be replicated in the host bacterium, it is not necessary to remove the tool plasmid. It should be noted that the use of suicide plasmid homologous recombination technology is a relatively safe and stable gene editing method that avoids the disruption of normal regulatory mechanisms.

## CONCLUSIONS

The overuse of antibiotics has led to the emergence of drug resistance in many common diseases. To understand drug resistance genes, researchers have investigated them from a functional genetic perspective. Bacteria use active efflux mechanisms, including energy-dependent efflux systems like quinolones, macrolides, and chloramphenicol, to reduce intracellular drug concentrations. These efflux systems can transport one or more classes of antibiotics. By knocking out or modifying these genes, researchers can explore changes in the resistance phenotype of knockout and proto-bacteria, laying the foundation for the development of modern drugs.

Drug transporter genes have been extensively studied through the sequencing of bacterial genomes. In *E. coli*, 37 open reading frames (ORF) have been identified as drug transport genes (*Paulsen, Sliwinski & Saier, 1998*), based on sequence similarity. However, there are still some drug resistance genes that have yet to be discovered. To understand bacterial gene function, researchers can use a variety of methods such as knockout and knock-in experiments, bioinformatics prediction, and other means to cross-check the function of genes. The prevalence of bacterial multidrug resistance is extensive across various bacterial species in nature. This article employs three knockout methods that utilize antibiotic resistance for screening knockout strains. These methods are commonly used in *E. coli*, particularly in strains with lower levels of antibiotic resistance. However, in the case of multi-drug resistant strains, selecting resistance genes for screening may pose challenges.

Knockout technologies have undergone three generations of evolution and continue to evolve due to their irreplaceable benefits in gene deletion. However, it should be noted that the plasmids used in these methods may not be stable in all strains, and therefore, knockout plasmids should be designed based on the characteristics of the corresponding strains.

## ACKNOWLEDGEMENTS

We express our gratitude to all those who have contributed to the writing of this review, as well as to the reviewers who participated in the publication process.

### Funding

This work was supported by the Key Research and Development Project of Heilongjiang Province of China (Grant No. GZ20210101), the Cultivation Project of Heilongjiang Bayi Agricultural University (Grant No. XDB-2016-22), and the Postdoctoral Scientific Research Start-up Fund of Heilongjiang (Grant No. LBH-Q21158). The funders had no role in study design, data collection and analysis, decision to publish, or preparation of the manuscript.

## Grant Disclosures

The following grant information was disclosed by the authors:
Heilongjiang Province of China: GZ20210101.
Heilongjiang Bayi Agricultural University: XDB-2016-22.
Heilongjiang: LBH-Q21158.

## Competing Interests

The authors declare that they have no competing interests.

## Author Contributions

- Chunyu Tong conceived and designed the experiments, authored or reviewed drafts of the article, and approved the final draft.
- Yimin Liang performed the experiments, prepared figures and/or tables, and approved the final draft.
- Zhelin Zhang analyzed the data, prepared figures and/or tables, and approved the final draft.
- Sen Wang analyzed the data, prepared figures and/or tables, and approved the final draft.
- Xiaohui Zheng analyzed the data, prepared figures and/or tables, and approved the final draft.
- Qi Liu analyzed the data, prepared figures and/or tables, and approved the final draft.
- Bocui Song conceived and designed the experiments, authored or reviewed drafts of the article, and approved the final draft.

## Data Availability

This is a literature review and did not utilize raw data.

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
