# Peer review of "Review of knockout technology approaches in bacterial drug resistance research"

_PeerJ, doi:10.7717/peerj.15790_

## Round 0.1 · original submission · Major Revisions

Two experts have reviewed the paper and present their criticisms below. Both reviewers note extensive language issues - please have your manuscript proofread by a proficient speaker of English.

Reviewer #1 points out that several similar reviews have been published in recent years. Novelty as such is not a requirement for PeerJ, but the authors should make clear their aims and any advances/differences in approach compared with other recent reviews. Reviewer #1 raises a concern about Figure 1. The figure is sufficiently changed to avoid plagiarism, but an acknowledgment of the inspiration for the figure (e.g., "adapted from") should be included.

Reviewer #2 notes that some aims mentioned in the abstract are not included in the paper itself. They also raise a recurrent issue of missing citations to bolster arguments.

Please respond to these and the other criticisms raised by the reviewers in a revised manuscript.

Reviewer 1 ·

Basic reporting

I believe the author’s intention is to provide a summary of the up-to-date methods to target (and potentially remove) genes that confer antimicrobial resistance.

The introduction needs much more detail. As is, it is very Spartan and does not go into depth enough. There is no obvious connection and no apparent reason for the review is provided.

Taking the reviewing guidelines in order, and to start with the language; there are multiple instances that the text was confusing to me for example
line 364 “Moreover, E. coli [sic], a model strain among bacteria, should not be anti to any antibiotics.” or
lines 317-318 “…splice transfer…” ,
line 162 “exons” and
line 174 “…phage arabinose promoter”.
line 351 "...righteous strand..."
line 361 "...plasmid shearing FRT sites."
line 372 "...methods exhibit several defects,..."

At other points the text appears very colloquial for example
lines 128 “Screening + reverse screening…”, 133 “(3) Screening + site-specific recombination…”, or contains plain statements such as
lines 359-363 which do not necessarily belong to a review manuscript.

Throughout the text there are additionally typographic errors such as
line 167 “Murers et al.(Muyrers et al., 1999)…”, or
lines 146-147 – bacterial species should be in italic
(more on referencing appropriately, in the study design)

The above mentioned examples are not the entirety of similar issues that created unnecessary confusion and complication while reading the text which to me designates the overall lack of attention to detail.

However my biggest concern is that there are several related published reviews for example https://www.mdpi.com/1424-8247/15/12/1498 ,
https://jnanobiotechnology.biomedcentral.com/articles/10.1186/s12951-021-01132-8 ,
https://www.ncbi.nlm.nih.gov/pmc/articles/PMC9356603/ ,
https://www.tandfonline.com/doi/full/10.1080/14787210.2017.1400379 .
The relative amount of information provided to the relevant topic at hand is minimal, compared to the extensive description of comparison of genetic knockout methods.

Overall I have the impression that this manuscript is not delivering what was intended (at least not to the extend I expected when I read the title) in terms of discussing in depth in the introduction section and the description/application section, both the drug resistance problem and its solution using genomic knock-outs.

Experimental design

The content should be within the aims and scope of the journal - falling under the "biological sciences" category.

The investigation/survey was a Pubmed database interrogation. The technical and ethical standards merit perhaps is not entirely relevant here. One could use the exact search terms and repeat the survey.
The terms ("gene knockout" , "bacterial drug resistance", "lambda-Red homologous recombination", "CRISPR/Cas9" and "suicide plasmid") used for the survey should result in the desired articles - if that was the intention of the authors.

In terms of quoting sources/paraphrasing in an appropriate way:
, fig.1 appears to be taken 1:1 from its original publication https://www.pnas.org/doi/10.1073/pnas.120163297 which is not explicitly mentioned neither in the figure legend nor in the text. I would expect to see here something like “adapted from” or “with permission” if the figure is so identical to the original.

For the claim of line 335, regarding the 37 ORFs, there is no reference given.

Validity of the findings

The layout of the conclusion section provided is equally confusing to me, mentioning yet again another set of methods (lines 338 and onward mentioning gene expression profiling, gene knock-in, gene silencing...) which I am not sure they belong in the conclusion section.

As mentioned above, lines 357-363, are not appropriate for the review context.

Reviewer 2 ·

Basic reporting

In this paper, Tong et al. present a review of gene knockout technologies in bacteria, with a focus on Red recombineering, CRISPR/Cas and suicide vectors. The review is suitably broad in context (i.e. methodology covering almost all bacterial species). The area does not seem to have been reviewed recently except for an abundance of CRISPR-specific publications.

Largely the text reads well but there are some language errors that could be fixed prior to publication. Proofreading by a native-level English speaker is recommended.

Typographical comments:
Line 89: LoxP should be capitalised (in Cre-LoxP)
Line 99-100: the phrase “during two recombination and complete recombination” is unclear. Is there a word missing here?
Line 108: “Systems” does not need to be capitalised.
Line 116: “has not been rarely reported” – should this read “has been rarely reported”?
Line 118: “then” should read “than”
Line 139: “has significantly” should read “has been significantly”
Line 146: Escherichia coli has already been mentioned at this point in the text, therefore it is unnecessary to write “Escherichia coli (E. coli)” here.
Line 162: “Exons encode exonucleases” should read “Exo proteins are exonucleases”
Line 163: Typo: “exons” again.
Line 165: “It prevents the degradation of linear dsDNA by Gam…” should read “Gam prevents the degradation of linear dsDNA by…”
Line 169: “Datsenko et al.” should read “Datsenko & Wanner”
Line 177: “Depending on the removal of the target gene” doesn’t make sense in this context.
Line 201: Comma instead of full stop after “II”.
Line 203: Random full stop in the middle of a sentence.
Line 213: should “are generally considered to be difficult repairing” instead read “are generally considered to struggle repairing”
Line 240: “is automatically” should read “is to be automatically”
Line 283: “Genegorging” should read “gene gorging”
Line 299: The abbreviation “PAM” has not been defined anywhere.
Line 300: The meaning of “DNA repair generally selects non-homologous recombination” is unclear.
Line 312: “can be performed without trace knockout” needs to be rephrased.
Line 317: The meaning of “after splice transfer” is unclear.
Line 327: “Bacteria reduce the concentration of drugs in bacteria” needs to be rephrased.
Line 330: What do the authors mean by “goes against” here?
Throughout: Bacterial names and gene names should be italicised.
Throughout: In many places, the authors of a publication are named and then these names duplicated in the citation immediately afterwards. This looks untidy. The citation should be date only, e.g. “Murers et al.(Muyrers et al., 1999)” could instead read “Muyrers et al. (1999)”.
Reference list: The authors have numbered their references whilst using full author-date citations in the main text. Please correct to one or the other according to the journal guidelines.

Experimental design

Study design is generally fine apart from one query:
Line 78-9: How do the authors define “high” impact factor and what is their specific rationale for using impact factor as a criterion in this review?

Validity of the findings

The review is largely complete but lacks important citations in some places, especially when the authors are making a seemingly subjective judgement (see specific comments below).

In the abstract, the authors claim to “propose a method to achieve rapid and efficient knockout”. No such method is really proposed in the text; it’s simply a review of existing methods and their perceived flaws (which are sometimes not backed up by any citations). Again there are important references missing in several places in the review, as noted below.

Line 118: Can the authors cite a study to reinforce the statement that “The main problem is off-target effects”?
Line 140: Can the authors please expand on “additional DNA experimental techniques”? This is very vague.
Lines 150-152: The authors should cite example publications where these problems have been demonstrated in the literature.
Line 170: I don’t think a plasmid can be auxotrophic per se.
Line 174: “The phage arabinose promoter” doesn’t make sense. The arabinose-inducible araBAD promoter is bacterial.
Line 176: “flip-flop binding sites” is incorrect. FRT stands for “flippase recognition target” so the authors should use this terminology.
Line 205: is “secreting” the right word for the mechanism here?
Line 237: Not all mutations effect the chromosome. Suicide vectors (and recombineering) can be used to modify other molecules such as large, low-copy plasmids.
Line 244: Suicide vectors can be transformed. They don’t always require conjugation.
Line 282: “Extremely low” is a huge overstatement. This technique has been used to generate tens of thousands of KO strains! If the authors are going to claim things like this, they need to cite actual numbers…
Line 307: Again, citation needed for these claims.
Line 364-5: “should not be anti to any antibiotics”… do you mean to say “resistant”? Unclear.
Conclusion section: The authors talk a lot about drug resistance, which isn’t really a conclusion to the rest of the review. Similarly, a completely new topic is introduced here in some detail (gene silencing), which is similarly not a conclusion. These aspects should be presented elsewhere in the review.

---

## Round 0.2 · accepted · Accept

The concerns of both reviewers have been addressed in the revised manuscript. as Reviewer #2 notes, there are still some typos and other small errors. However, these can be corrected at the proofing stage. Please read the proofs carefully to correct any remaining typos.

Reviewer 2 ·

Basic reporting

The revised text is greatly improved by the authors in terms of both language and content. However, there are some very minor typographical errors that remain (e.g. italics need to be used for gene names such as ompK36). Fixing these could improve the manuscript prior to final publication.

Experimental design

The authors have clarified their study design and I have no further queries/issues.

Validity of the findings

The authors have improved their references and interpretation, making it far more robust.

Additional comments

Overall the authors have made a large number of changes to the text that have resulted in a much improved manuscript, and I believe it is now suitable for publication.